# Analyzing Cognitive Plausibility of Subword Tokenization

**Lisa Beinborn**
CLTL Lab
Vrije Universiteit Amsterdam
Amsterdam, The Netherlands
l.beinborn@vu.nl

**Yuval Pinter**
Department of Computer Science
Ben-Gurion University of the Negev
Beer Sheva, Israel
uvp@cs.bgu.ac.il

## Abstract

Subword tokenization has become the de-facto standard for tokenization, although comparative evaluations of subword vocabulary quality across languages are scarce. Existing evaluation studies focus on the effect of a tokenization algorithm on the performance in downstream tasks, or on engineering criteria such as the compression rate. We present a new evaluation paradigm that focuses on the cognitive plausibility of subword tokenization. We analyze the correlation of the tokenizer output with the response time and accuracy of human performance on a lexical decision task. We compare three tokenization algorithms across several languages and vocabulary sizes. Our results indicate that the UnigramLM algorithm yields less cognitively plausible tokenization behavior and a worse coverage of derivational morphemes, in contrast with prior work.

## 1 Introduction

When we develop natural language processing (NLP) models, we first need to segment a stream of text into small processable units. This preparatory step is known as **tokenization** and it is more challenging than the segmentation of continuous sensor signals because human language uses symbolic representations. Traditionally, the space-delimited word was considered a meaningful basic unit, but the concept cannot intuitively be mapped to languages with a rich morphological structure such as Turkish or Finnish, or to languages that use a writing system without white spaces such as Chinese.

More recently, the focus has shifted to smaller character sequences known as **subwords**, with the explicit goal of limiting the necessary vocabulary size (which affects model size and performance), and the implicit hope of better approximating semantically-meaningful linguistic units below the word level, i.e., morphemes. In practice, today's dominant subword tokenization algorithms are purely data-driven. They treat frequent

| Sequence | Tokens | CHUNK | RT | Acc |
|---|---|---|---|---|
| seafood | seafood | 0.86 | 578 | 0.97 |
| outfoxed | out-fo-x-ed | 0.50 | 734 | 0.62 |
| *brithbloom | br-ith-blo-om | 0.60 | 693 | 0.97 |
| *catchwind | catch-wind | 0.78 | 788 | 0.82 |

Table 1: Examples of lexical decision stimuli, the average response time (RT) and accuracy (Acc) of human responses, and the output of a WordPiece tokenizer with a vocabulary size of 50,000. Non-words are marked with an asterisk (*). Chunkability (CHUNK) is calculated based on the rate of tokens per character (1).

sequences as single tokens (e.g., seafood), and split less frequent ones into multiple tokens composed of frequently occurring character sequences (e.g., seabirds → seab-ird-s). Subword splits might coincide with morpheme boundaries (the plural marker s), but not necessarily (seab).

Previous comparisons of tokenization algorithms focused on engineering-oriented desiderata such as processing speed and encoding efficiency, and on the performance of models on downstream NLP tasks (Rust et al., 2021). In this paper, we evaluate subword tokenizers from a cognitive perspective, utilizing lexical decision task measures as a proxy for the processing complexity of individual words. We evaluate the split rates of three tokenization algorithms on various languages. We find significant correlations in line with cognitive expectations, allowing systematic analyses of the influence of parameters such as vocabulary size. We observe that the UnigramLM tokenization algorithm (Kudo, 2018) produces less correlative splits than BPE (Sennrich et al., 2016) and Word-Piece (Schuster and Nakajima, 2012), in contrast with previous evaluations made over corpus statistics and downstream tasks (Bostrom and Durrett, 2020). In further experiments, we find that multilingual token vocabularies inhibit tokenizers' ability to predict cognitive performance as well as signs that current popular vocabulary sizes are insuffi-

cient for morphologically-rich languages, echoing recent findings (Liang et al., 2023).[1]

## 2 Self-Supervised Tokenization

A tokenizer takes as input a sequence $S$ of characters $[c_1, \ldots, c_n]$ and splits it into non-overlapping substrings, the tokens $[t_1, \ldots, t_k]$, where $k \leq n$. Each token $t_i$ consists of a variable number of $j$ consecutive characters ($1 \leq j \leq n$), such that the concatenation of the tokens $t_i$ yields the sequence $S$. A tokenizer consists of a vocabulary consisting of $m$ tokens and an algorithm that determines the best splits of the input $S$ into vocabulary items $t_i$. See Mielke et al. (2021) for a detailed survey of tokenization approaches.

**Evaluating Vocabularies** Comparative evaluations of tokenization algorithms commonly focus on downstream performance and on cross-lingual differences. Maronikolakis et al. (2021) calculate the tokenization compatibility for pairs of languages and find that the vocabulary size of a tokenizer needs to be adapted to the characteristics of the language. Multilingual language models use a single shared vocabulary for a large number of languages to facilitate cross-lingual transfer, however, Rust et al. (2021) find improvements when these are replaced with targeted monolingual tokenizers. Liang et al. (2023) propose to increase the vocabulary size for multilingual models and assign per-language budgets in a dynamic manner, in order to mitigate effects on the splitting ratio for languages less represented in the vocabularies. They de-emphasize token sharing between languages with little lexical overlap, in line with Chung et al. (2020). Yehezkel and Pinter (2023) propose to incorporate context sensitivity in order to generate more cohesive tokenization and show that their approach leads to increased downstream performance for both English, and the morphologically more complex language Turkish.

**Morphological Evaluation** More linguistically motivated evaluations of subword tokenization focus on morphological plausibility. Bostrom and Durrett (2020) compare the BPE and UnigramLM algorithms for English and Japanese and find that the segmentation produced by the latter aligns more closely with morphology and leads to better results on downstream tasks, especially for Japanese. In

a similar vein, Park et al. (2021) find that BPE does not properly reflect morphological complexity and that enriching the model with explicit morphological information leads to reduced language modeling surprisal. Hofmann et al. (2022) show that a vocabulary with better morphological coverage leads to better performance in genre classification of English titles, and might lead to better generalization capabilities (Hofmann et al., 2021). Other studies have shown that these consistent results in English do not necessarily generalize to other languages (Mager et al., 2022), particularly morphologically-rich ones (Klein and Tsarfaty, 2020).

Morphological segmentation is related to tokenization, but it is sensitive to phonotactic variations, e.g., `discernible` is segmented into `discern` and `-able` (Batsuren et al., 2022). Nevertheless, the winning system at the SIGMORPHON shared task was based on subword tokenization and outperformed character-based approaches, indicating that subwords can approximate morphological boundaries (Peters and Martins, 2022).

**Cognitive Plausibility** From a cognitive perspective, it remains an open question to which extent lexical processing is driven by morphological units. One of the most robust effects in lexical decision tasks (Amenta and Crepaldi, 2012) is that morphologically structured non-words cause longer response times and lead to decreased accuracy in word detection. Beyersmann et al. (2020) find that this effect is stronger for German than for French, and suggest that this is due to its larger degree of morphological productivity. Dawson et al. (2021) compare lexical decision times for English words and find that priming with morphological components (`teach`) leads to faster responses (`teacher`) even if the prime has no semantic relation (`corn` → `corner`). (Yang et al., 2022) show that the prediction of eye fixations is facilitated for English and Dutch readers by operating on subtokens determined by unsupervised tokenizers instead of word units. Stevens and Plaut (2022) claim that effects attributed to morphological decomposition cannot be easily disentangled from frequency effects, and urge NLP researchers to integrate response times into the evaluation of distributional approaches.

## 3 Experimental Setup

We train three tokenization algorithms on 100,000 sentences from the news domain of the Leipzig

---

[1] All analyses are available on github: `https://github.com/clap-lab/cogtok`

| Language | Participants | Words | Non-Words |
|----------|-------------|-------|-----------|
| English | 78 | 28,730 | 27,137 |
| Dutch | 81 | 14,089 | 14,089 |
| French | 975 | 38,335 | 38,807 |
| Spanish | 209,351 | 45,223 | 56,861 |

Table 2: Summary statistics of the lexical decision data.

corpus (Biemann et al., 2007),[2] and introduce the chunkability metric to evaluate tokenizer output against cognitive data from a lexical decision task.

**Tokenization Models** We use the Huggingface implementations of three corpus-based subword algorithms: byte-pair encoding (BPE), WordPiece (WPC), and Unigram (UNI).[3]

**BPE** originated as a compression algorithm (Gage, 1994; Sennrich et al., 2016) and has been used in large pre-trained language models such as GPT-2 (Radford et al., 2019). BPE vocabularies are built bottom-up, starting with an initial vocabulary of all characters. The algorithm then iteratively merges sequences of characters frequent in the corpus and adds them as tokens to the vocabulary until reaching the maximum size. During inference, an input sequence is greedily split into tokens aiming for a minimum number of splits, see Gallé (2019) for a more detailed description. **WordPiece** (Schuster and Nakajima, 2012) is a variant of BPE which adds tokens to the vocabulary when they maximally increase the likelihood of an n-gram-based language model in the corpus, and is decoded greedily left-to-right to find the locally longest token available at each step. The **UnigramLM** (Kudo, 2018) algorithm, in contrast, takes a top-down approach starting with an overly large vocabulary of all possible tokens, followed by iteratively pruning those that lead to minimal loss of likelihood over the corpus when they are removed from a token-unigram language model.

**Cognitive Data** We use data from lexical decision tasks in British English (Keuleers et al., 2012), Dutch (Keuleers et al., 2010), French (Ferrand et al., 2010), and Spanish (Aguasvivas et al., 2018). In these tasks, participants are presented with a sequence of characters, e.g., thornier, and decide whether the sequence forms a valid word in

their first language. The datasets contain information about the average *response time* (i.e., the number of milliseconds it took the participants to make a decision)[4] and accuracy for each stimulus. Table 2 provides an overview of the number of participants and stimuli for each dataset. Each participant only saw a subset of the stimuli; further details about the data collection are available in the original references. Since the Spanish study was a crowd-sourcing project, we removed outliers with a reported response time in the first and last percentiles ($<484$ and $>7,753$ ms, respectively).

**Metric** A sequence of characters $[c_1, \ldots, c_n]$ is split into tokens $[t_1, \ldots, t_k]$. We base our metric on the intuition that fewer splits generally indicate a better fit and that longer sequences are more likely to be split. Our metric therefore takes both $n$ and $k$ into account to measure how well the tokenizer handles the sequence. We define the **chunkability** of a sequence as a value that decreases as the ratio of tokens over characters increases:

$$\text{chunkability} = 1 - \frac{\#\text{tokens}}{\#\text{chars}} = 1 - \frac{k}{n}. \quad (1)$$

If a token is split into individual characters, chunkability is zero. For tokens that are not split, chunkability approaches one as the number of characters increases, reflecting the increasing challenge of fitting long words into the vocabulary.[5]

## 4 Results

We test the hypothesis that sequences with higher chunkability are easier to process for humans and are more likely to be considered words. Figure 1 visualizes Pearson's correlation between the chunkability of a sequence and the response time and accuracy observed from humans rating the sequence, and Table 1 illustrates some typical examples. We can see that for a sequence that qualifies as a word, like seafood, a higher chunkability score (i.e., easier processing by the tokenizer) is likely to co-occur with higher accuracy and a lower response time. For non-words, we observe the reverse tendencies: non-word sequences with a high chunkability such as catchwind require longer response times and

---

[2]In pilot experiments, we explored larger training sizes but did not observe relevant differences.

[3]https://github.com/huggingface/tokenizers. We focus on character-segment models, while other approaches operate on the single character, byte, or pixel level (Clark et al., 2022; Xue et al., 2022; Rust et al., 2023).

[4]While this duration is usually characterized as *reading time* in the resources, we agree with the observation of a reviewer that it remains unclear how much time the participants spend reading and use the term *response time* instead.

[5]For comparison, we also ran our experiments by using the number of splits (without normalization) as our metric, see Appendix B.

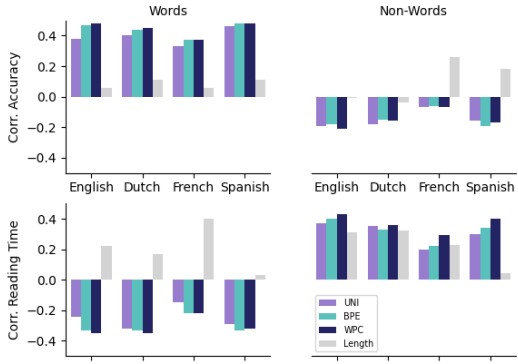

Figure 1: Pearson correlation between the chunkability of a sequence and the observed accuracy and response time of human participants in a lexical decision task. All tokenizers are trained on the same data (per language) with a vocabulary size of 50,000. In all cases, model results are significantly different from the length baseline ($p<0.01$) according to the Fisher z-transformation (using an implementation by Philipp Singer: https://github.com/psinger/CorrelationStats). The comparisons between UNI and BPE/WPC are significant in 22 out of 32 conditions. The differences between BPE and WPC are not significant in most comparisons.

are less accurately identified as non-words than an unusual sequence such as brithbloom. These patterns are consistent across algorithms and languages, whereas a baseline that only considers character length cannot capture the effect (longer sequences generally lead to longer response times). The correlation for the UnigramLM algorithm is systematically lower than for the other two algorithms, suggesting a contrast with morphology- and corpus-based measures considered by Bostrom and Durrett (2020). For French, chunkability seems to be less correlated with human responses, echoing findings related to cognitive performance in a morphologically-primed environment (Beyersmann et al., 2020).

While the absolute correlation scores might be of limited explanatory value, we find the relative differences between conditions a relevant point of information for further development.[6]

[6]For a complementary perspective, we ran a linear regression analysis on the words to compare to a frequency measure which can be found in Appendix C. We are also considering alternative correlation metrics such as Spearman's, Kendall's, and Goodman-Kruskal. Initial analyses indicate that they capture similar tendencies.

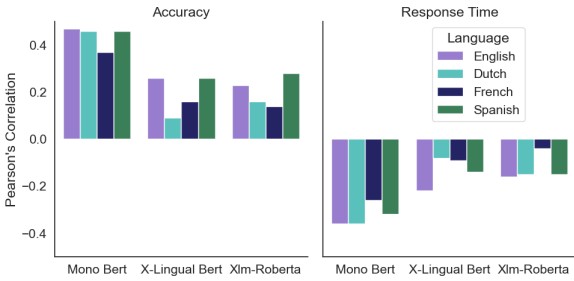

Figure 2: Differences between pre-trained monolingual and multilingual tokenizers measured as Pearson correlation between the chunkability of words and the observed accuracy and response time of human participants in a lexical decision task. All differences between the tokenizers' performance are statistically significant ($p<0.01$), except for two comparisons of the multilingual tokenizers (correlation with accuracy for French and correlation with reading time for Spanish).

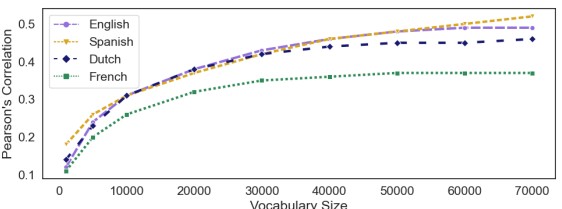

Figure 3: Pearson correlation between the WordPiece chunkability of a word and the observed accuracy in human responses to a lexical decision task, as vocabulary size grows.

**Cross-lingual Vocabulary** It has been shown that the "curse of multilinguality" reduces the performance of multilingual models compared to their monolingual counterparts (Conneau et al., 2020). Rust et al. (2021) show that this difference in performance is strongly related to the tokenizer. We compare cognitive plausibility of monolingual and cross-lingual tokenizers of pretrained models in Figure 2, and affirm that the monolingual tokenizer aligns much better with human responses than the cross-lingual ones.[7] See also Appendix D.

**Vocabulary Size and Morphology** The chunkability values vary with the size of the vocabulary of the tokenizers. Figure 3 shows how the correlation with human responses increases with the vocabulary size of the WordPiece tokenizer until

[7]We use the following huggingface models: GroNLP/bert-base-dutch-cased (Dutch), bert-base-uncased (English), camembert-base (French), dccuchile/bert-base-spanish-wwm-uncased (Spanish), bert-base-multilingual-uncased and xlm-roberta-base (crosslingual). BERT-based models use WordPiece tokenization, XLM-RoBERTa uses BPE.

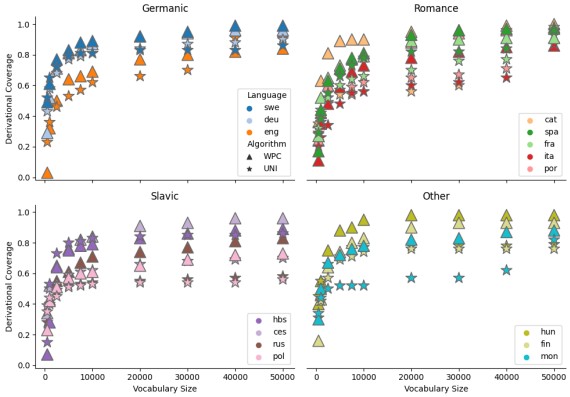

Figure 4: Coverage of derivational morphemes in the vocabulary of WordPiece ($\triangle$) and UnigramLM ($\star$) for 13 languages.

it plateaus for most languages at 50,000.[8] To test whether this effect may be related to morphological coverage, we use morphological annotations for 13 languages from the SIGMORPHON shared task (Batsuren et al., 2022) and extract an inventory of derivational morphemes for each language. We only used derivational morphemes which occur in at least 0.1% of the annotations (to avoid rare morphemes such as oculo), yielding between 80 and 140 derivational morphemes per language. From the coverage curves in Figure 4, we see that derivational coverage increases with vocabulary size, but morphologically-rich languages such as Russian, Polish, and Mongolian remain unsaturated even with a vocabulary size of 50,000. This suggests that previous work on morphological segmentation which used significantly smaller vocabulary sizes (e.g., Peters and Martins, 2022) did not uncover the full potential of the approach. We also see that WordPiece tokens overall provide better coverage of morphemes than UnigramLM, reinforcing our findings from the previous experiments.

## 5   Conclusion

We propose a new evaluation paradigm for comparing subword tokenization algorithms using cognitive data. We introduce a novel metric to capture the chunkability of a sequence that correlates with cognitive phenomena of lexical recognition. The overall trends suggest that the connection between plausibility tasks and segmentation is meaningful enough to be used as a benchmark. We find a lower cognitive correlation for the UnigramLM algorithm than for WordPiece and BPE, which does not nec-

---

[8]The tendencies for BPE and UNI are comparable.

essarily align with previous work evaluating tokenizers on morphological segmentation and downstream performance, suggesting that our framework provides a complementary perspective to tokenizer vocabulary evaluation. Our analyses on vocabulary size and morphological coverage provide initial insights towards the development of cognitively and linguistically more plausible tokenizers.

## Limitations

Our cognitive analyses are limited to two Romance and two Germanic languages. The response times were collected as separate experiments with slight variations in the data collection procedure (i.e., number of stimuli per participant, background of participants) and might not be directly comparable. We average over the responses, which may conceal individual differences between respondents (Plank et al., 2014; Kidd et al., 2018; Pavlick and Kwiatkowski, 2019). Pearson's $\rho$ has a tendency to pick up spurious correlations (Aldrich, 1995), which is why we abstract from absolute values and focus on relative differences between conditions. The quality of the selection of derivational morphemes is determined by the characteristics of the SIGMORPHON datatset.

## Ethics Statement

We use datasets that have been fully anonymized and adhere to ethical guidelines for data collection. Our analyses do not reveal metadata of the participants that would enable identification. Claims about cognitive plausibility need to be made with caution because the procedural patterns underlying human language processing still remain an open research question. We have therefore paid special attention to a realistic interpretation of our results and avoid overpromising messages (Lipton and Steinhardt, 2019).

## Acknowledgements

Lisa Beinborn's work was supported by the Dutch National Science Organisation (NWO) through the VENI program (Vl.Veni.211C.039).
Yuval Pinter's work was supported by a Google gift intended for work on *Meaningful Subword Text Tokenization*.
We thank the reviewers for their thoughtful comments. We thank Joshua Snell and Omri Uzan for comments on early drafts.

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

## A Language Codes

The abbreviations used in Figure 4 correspond to ISO 639-3 codes and are mapped as follows: swe = Swedish, deu= German, eng= English, cat= Catalan, spa= Spanish, fra= French, ita= Italian, por= Portuguese, hbs= Serbo-Croatian, ces= Czech, rus= Russian, pol= Polish, hun= Hungarian, fin= Finnish, mon= Mongolian.

## B Number of Splits

In the chunkability metric, we normalize by the length of the sequence. For comparison, we also ran our experiments by simply using the number of splits as a metric. The tendencies in the results in Figure 5 are comparable for the two metrics except for the correlations with the response time for non-words which are substantially less consistent with cognitive phenomena for the number-of-splits measure. We assume that this can be explained through the finer range of values afforded by chunkability.

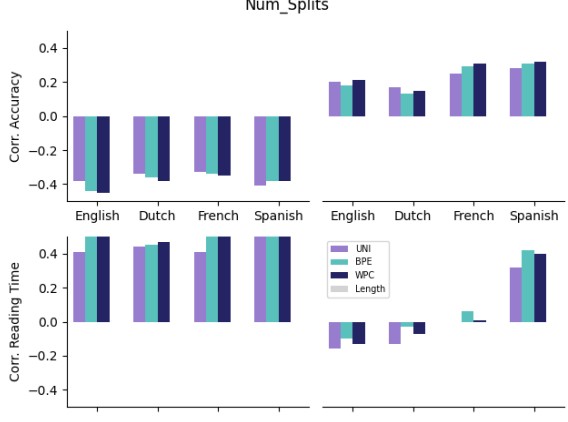

Figure 5: Pearson correlation between the number of splits a tokenizer assigns to a sequence and the observed accuracy (top) and response times (bottom) in human responses to a lexical decision task. All tokenizers are trained on the same data (per language) with a vocabulary size of 50,000. Note that the correlation directions are reversed compared to Figure 1, as we did not take the inverse.

## C Regression Analysis

In this paper, we focus on the evaluation of sub-word tokenizers and not on predicting cognitive phenomena. However, one of the reviewers inspired us to run a demonstrative linear regression analysis for the words in the dataset. For all four languages, we found that both chunkability and

frequency obtained similar low mean squared error (0.00–0.06) on the test data for both response time and accuracy. However, all explained variance scores are negative (with systematically higher scores when predicting with frequency and particularly low scores for French and Dutch response times). We assume that the low explained variance is related to the finegrained cognitive signal and to individual differences in the responses. Prediction would probably be easier when predicting classes (e.g., high/low) instead of absolute values. We are interested in diving deeper into these pilot analyses in cooperation with cognitive scientists. Finally, we note that while frequency is only available for true words, chunkability can be a proxy for frequency effects in non-words as well.

| Lang | Signal | MSE CHUNK | MSE FREQ | EV CHUNK | EV FREQ |
|------|--------|-----------|----------|----------|---------|
| eng | RT | .01 | .01 | -14.73 | -0.54 |
| eng | Acc | .06 | .05 | -3.73 | -0.91 |
| nld | RT | .03 | .03 | -38.28 | -2.51 |
| nld | Acc | .04 | .04 | -5.01 | -2.33 |
| fra | RT | .02 | .01 | -124.78 | -2.28 |
| fra | Acc | .02 | .02 | -6.96 | -4.24 |
| spa | RT | .01 | .00 | -18.18 | -0.03 |
| spa | Acc | .06 | .05 | -2.88 | -0.56 |

Table 3: Linear regression results as mean squared error (MSE) and explained variance (EV) for the words in our dataset with a random 80/20 train-test split. We compare the two features chunkability (CHUNK) and word frequency (FREQ). Frequency is determined using Zipf frequency scores obtained from the wordfreq package v3.03 (https://pypi.org/project/wordfreq), and chunkability is determined using the WordPiece tokenizer. We predict the two signals response time (RT) and accuracy (Acc) separately using the standard linear regression model from the sklearn package v1.3 (https://scikit-learn.org/stable). Both signals are normalized using min-max scaling.

## D Four-lingual Vocabulary

In order to better control the effect of vocabulary sharing, we also trained tokenizers on all the English, Dutch, French, and Spanish training data jointly. Figure 6 illustrates the results for the Word-Piece tokenizer and shows that the correlation is lower for multilingual models but improves when increasing the vocabulary.

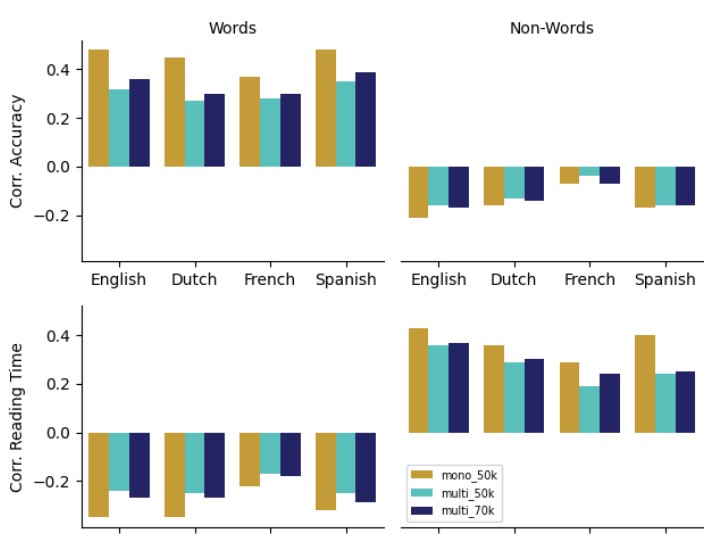

Figure 6: Pearson correlation between the chunkability of a sequence and the observed response times and accuracy in human responses to a lexical decision task. To determine the chunkability, we trained the WordPiece (WPC) tokenizer on monolingual and four-lingual training data with vocabulary sizes of 50k and 70k.