# OpenReview forum: "Analyzing Cognitive Plausibility of Subword Tokenization"
_EMNLP/2023/Conference — EMNLP 2023 Main_

### Official Review · Reviewer_6Wn8 · 2023-08-02

**Soundness:** 4

**Excitement:**

4: Strong: This paper deepens the understanding of some phenomenon or lowers the barriers to an existing research direction.

**Missing References:**

N/A

**Paper Topic And Main Contributions:**

This paper presents "chunkability" as a new evaluation paradigm for subword tokenization. This evaluation metric is meant to evaluate the cognitive plausibility of subword tokenizers as compared to humans' lexical decision times and accuracy, and correlations of these measures are presented for several popular subword tokenization algorithms.

**Questions For The Authors:**

As presented, it seems that the chunkability metric is affected by word length in ways that may not be desirable: does a 5-character word with one token really have a lower chunkability than a 10 character one with one token? Is this desirable, and if so, why? I wonder if the authors tried similar metrics that do not factor in length as explicitly? I'd like to see a couple of sentences discussing this.

**Reasons To Accept:**

I see this as an impactful short paper: it proposes a fundamental rethinking of how we analyze and evaluate subword tokenizers. I think the push to evaluate tokenizers in terms of cognitive plausibility is the most important contribution of the paper, above and beyond the specific method proposed, though I also see that method as being promising.

**Reasons To Reject:**

I'd like to see a more thorough analysis of the results along the following dimensions:
* In correlating with human metrics, the authors don't control (to my knowledge) for potential confounds such as frequency, which is known to affect lexical decision time. A regression analysis including such metrics would be more compelling; I would rate this paper significantly higher were one given and I think this is the most important point for revisions if the paper is accepted.
* In Figures 1 and 2, I'd like to see the y-axes rescaled either to -1 to 1 or -0.5 to 0.5 for a more consistent reading of the results; more generally, I'd like to see the authors acknowledge more explicitly that none of the correlations reported are particularly high (which is really fine to acknowledge in this setting, since their contribution is the evaluation metric). I'd also like to see significance testing where the authors report differences in correlation between models.
* The authors mention the shortcomings of Pearson's correlation in their limitations section, but I'd like to see another statistic presented because of this -- something like Goodman-Kruskall might be promising to report.

I think all of these can be easily addressed in the additional page, and aside from the first point, they do not detract from my overall excitement about the proposal.

I'd also like to see some discussion of the segmentations in Table 1: the proposed tokens for e.g. outfoxed to not correspond to the constituent morphemes of this word, and I'd like to see some discussion of this mismatch. If the subword tokenizer is often finding tokens that don't directly correspond to morphemes, then I think the cognitive plausibility claims are probably weaker. Again, though, this does not detract from the main aim of the paper, which is to provide a novel evaluation procedure.

**Reproducibility:**

4: Could mostly reproduce the results, but there may be some variation because of sample variance or minor variations in their interpretation of the protocol or method.

**Reviewer Confidence:**

4: Quite sure. I tried to check the important points carefully. It's unlikely, though conceivable, that I missed something that should affect my ratings.

**Typos Grammar Style And Presentation Improvements:**

I think the presentation of the paper could be improved by presenting chunkability earlier (i.e. before the start of S3, in its own section). In my view, this metric is the main contribution of the paper, and should thus be presented before the details of the experimental setup.

---

> ### Author Rebuttal · Authors · 2023-08-25
>
> We thank you for your constructive comments.
>
> > In correlating with human metrics, the authors don't control (to my knowledge) for potential confounds such as frequency, which is known to affect lexical decision time. A regression analysis including such metrics would be more compelling; I would rate this paper significantly higher were one given and I think this is the most important point for revisions if the paper is accepted.
>
> Answer:
> The original papers that present the lexical decision datasets provide analyses related to frequency. However, these analyses are limited to words, as non-words cannot be found in corpora. For non-words, only the frequency of smaller parts, e.g. character bigrams, can be measured. By design, the subword tokenizers considered in our experiments do just this: they **determine splits based on the frequencies of character sequences**. Using subword tokenizers is a more elegant way of accounting for non-word frequencies because the length of the character sequence under consideration does not need to be hard-coded but is determined by the sequences seen in the training data.
>
> In this paper, we focus on the evaluation of subword tokenizers and not on predicting cognitive phenomena. However, out of curiosity, we ran a small linear regression analysis comparing the predictive power of chunkability scores of the WordPiece tokenizer to Zipf frequency scores obtained from the wordfreq library for the words in the dataset. We split the data into 80% train and 20% test. For all four languages, we found that both chunkability and frequency obtain similar low mean squared error (between 0 and 0.06) on the test data for both reading time and accuracy. Overall, frequency yields better explained variance probably because it is a more finegrained signal. We are interested in diving deeper into these pilot analyses in cooperation with cognitive scientists.
>
> > In Figures 1 and 2, I'd like to see the y-axes rescaled either to -1 to 1 or -0.5 to 0.5 for a more consistent reading of the results; more generally, I'd like to see the authors acknowledge more explicitly that none of the correlations reported are particularly high (which is really fine to acknowledge in this setting, since their contribution is the evaluation metric).
>
> Answer:
> Thank you for pointing out the slight inconsistency in the negative limit of the y-axis scales of Figures 1 and 2. We will adjust this in the next version. Due to our large sample sizes, all reported correlations are significant (p<=0.01). Indeed, our work focuses more on relative differences between tokenizers. We report on significance of pair-wise comparisons below.
>
> > I'd also like to see significance testing where the authors report differences in correlation between models.
>
> Answer:
> We agree that significance calculations are a valuable addition to the paper.
>
> For the results in Figure 1, all comparisons to the length baseline are significant (p<=0.01). The comparisons of the unigram tokenizer to BPE and WPC are significant in 22 out of 32 conditions. The differences between the BPE and the WPC tokenizer are not significant in most comparisons.
>
> In Figure 2, 22 out of the 24 comparisons are significant. The two insignificant differences correspond to comparisons between the two multilingual tokenizers.
>
> These observations support the conclusions presented in the paper and we will add them to the next version.
>
> > The authors mention the shortcomings of Pearson's correlation in their limitations section, but I'd like to see another statistic presented because of this -- something like Goodman-Kruskall might be promising to report.
>
> Answer: We understand your reservation and will consider alternative metrics. Due to the large number of participants in the cognitive benchmarks, and the well-known effects of individual variability, we are not sure a rank-based correlation metric such as Goodman-Kruskall would be suitable for comparing against a comparatively coarse-grained chunkability score.
>
> > I'd also like to see some discussion of the segmentations in Table 1: the proposed tokens for e.g. outfoxed do not correspond to the constituent morphemes of this word, and I'd like to see some discussion of this mismatch. If the subword tokenizer is often finding tokens that don't directly correspond to morphemes, then I think the cognitive plausibility claims are probably weaker. Again, though, this does not detract from the main aim of the paper, which is to provide a novel evaluation procedure.
>
> Answer: This is a very relevant observation. Recent related work indeed found that subword splits do not always align with morpheme boundaries especially when the vocabulary size is small (e.g., [1], see also Figure 4 in our paper). It has been speculated that the subword tokenization algorithms are overfitted to English and are less useful for morphologically richer languages [2--3]. The relationship between morphological plausibility, performance, and cognitive plausibility remains an open research question and we think that our proposed methodology provides a useful tool for deeper analyses.
>
> [1] Valentin Hofmann, Janet Pierrehumbert, and Hinrich Schütze. ACL 2021. Superbizarre Is Not Superb: Derivational Morphology Improves BERT’s Interpretation of Complex Words
>
> [2] Stav Klein and Reut Tsarfaty. SIGMORPHON 2020. Getting the ##life out of living: How Adequate Are Word-Pieces for Modelling Complex Morphology?
>
> [3] Manuel Mager, Arturo Oncevay, Elisabeth Mager, Katharina Kann, and Thang Vu. ACL-findings 2022. BPE vs. Morphological Segmentation: A Case Study on Machine Translation of Four Polysynthetic Languages.
>
> > As presented, it seems that the chunkability metric is affected by word length in ways that may not be desirable: does a 5-character word with one token really have a lower chunkability than a 10 character one with one token? Is this desirable, and if so, why? I wonder if the authors tried similar metrics that do not factor in length as explicitly? I'd like to see a couple of sentences discussing this.
>
> Answer: This is a very good question which we also discussed when designing the experimental setup. We ran comparative analyses using only the number of splits without controlling for length but had not added them to the paper due to space limitations. We will add these figures to the appendix for comparison and use the extra page for additional explanations. The tendencies in the results are comparable for the two metrics except for the correlations with the reading time for non-words which are substantially less consistent with cognitive phenomena for the number-of-splits measure. This can be explained through the finer range of values afforded by chunkability. When comparing a 5-character word with a 10-character one, we indeed believe that the latter should present slightly higher chunkability as it is both more challenging for a tokenizer to incorporate it as a single token in its vocabulary and less likely a-priori to be a valid word.
>
> > Typos Grammar Style And Presentation Improvements: I think the presentation of the paper could be improved by presenting chunkability earlier (i.e. before the start of S3, in its own section). In my view, this metric is the main contribution of the paper, and should thus be presented before the details of the experimental setup.
>
> Answer: The introduction of the new metric at the end of the experimental setup is indeed not ideal. Adding a section before S3 seems like a promising alternative when given an additional page and we will consider it.

---

### Official Review · Reviewer_1agx · 2023-08-03

**Soundness:** 3

**Excitement:**

4: Strong: This paper deepens the understanding of some phenomenon or lowers the barriers to an existing research direction.

**Missing References:**

To the best pf my knowledge, all the most relevant papers are cited.

**Paper Topic And Main Contributions:**

The short paper focuses on subword tokenization algorithms by comparing their performance with human lexical decision results, in a few languages (two germanic and two romance ones, i.e. German, English and Spanish, French).

**Questions For The Authors:**

Sometimes you write “reading time” referring to “reaction time”:
are you sure that the two cognitive processes may be considered as overlapping or as the same?

At the beginning of the Results section, indirectly, you define “chunkability” as a measure of “wordlikeness”: are you sure of that? Do you think that the two properties can be considered at the same level, invoking the same cognitive processes?


**Reasons To Accept:**

The research rationale is well defined and rises an important question: how cognitive plausible are tokenizer algorithm results?
Results suggest a new evaluation paradigm that deserves an open discussion with the conference community

**Reasons To Reject:**

No reason to reject it

**Reproducibility:**

3: Could reproduce the results with some difficulty. The settings of parameters are underspecified or subjectively determined; the training/evaluation data are not widely available.

**Reviewer Confidence:**

4: Quite sure. I tried to check the important points carefully. It's unlikely, though conceivable, that I missed something that should affect my ratings.

**Typos Grammar Style And Presentation Improvements:**

My suggestion is to change those issues related to my questions to the author(s).

---

> ### Author Rebuttal · Authors · 2023-08-25
>
> We thank you for your constructive comments.
>
> > Sometimes you write “reading time” referring to “reaction time”: are you sure that the two cognitive processes may be considered as overlapping or as the same?
>
> Answer:  We used the term “reading time” in the text because it was used by the creators of the datasets. We indeed don’t know how much time the participants spend on reading which is why we added the clarification in parentheses in lines 194ff: “The datasets contain information about the average reading time (i.e., the number of milliseconds it took the participants to make a decision) and accuracy for each stimulus.” In order to make this point more clear, we will update the text to consistently use the term “response time” instead.
>
> > At the beginning of the Results section, indirectly, you define “chunkability” as a measure of “wordlikeness”: are you sure of that? Do you think that the two properties can be considered at the same level, invoking the same cognitive processes?
>
> Answer: Thank you for pointing out our slightly imprecise language. Our analyses are meant to find a relationship between the chunkability assigned by the subword tokenizer and the human reactions. Based on the finding that higher chunkability correlates with longer reading times and lower accuracy for non-words, we assume that high chunkability indicates that participants are more likely to consider the sequence to be a word, but further research is necessary to identify whether this assumption holds. We will phrase our findings more carefully.

---

### Official Review · Reviewer_ye9p · 2023-08-04

**Soundness:** 3

**Excitement:**

3: Ambivalent: It has merits (e.g., it reports state-of-the-art results, the idea is nice), but there are key weaknesses (e.g., it describes incremental work), and it can significantly benefit from another round of revision. However, I won't object to accepting it if my co-reviewers champion it.

**Paper Topic And Main Contributions:**

This paper adresses  the question of subword tokenization in the perspective of its cognitive plausibility. It is based on the study of the classical subword tokenization techniques (Byte-pair encoding, WordPiece and Unigram) and the correlation of their outputs with the reading time. The hypothesis is that morphologically structured non-words cause longer reading times and lead to decreased accuracy in word detection. The proposed metric is based on chunkability (the inverse of the ratio of tokens over characters) with the idea that  chunkability level can be correlated with processing difficulty: higher chunkability corresponds to an easier process, higher accuracy and lower reading times. Results show a better prediction with BPE and WPC.

**Reasons To Accept:**

This paper is clear and well-written. The cognitive perspective introduced in the task is interesting and provides new lights on the subword tokenisation mechanism. This technique is of course efficient, but nothing indicates that it is used by the human parser. These results could argue in favor of the idea that input segmentation is based on low-level techniques, delaying (or even avoiding in some cases) a lexical access to the mental lexicon.

**Reasons To Reject:**

The methodology is not original and the confirm known results.

**Reproducibility:**

4: Could mostly reproduce the results, but there may be some variation because of sample variance or minor variations in their interpretation of the protocol or method.

**Reviewer Confidence:**

4: Quite sure. I tried to check the important points carefully. It's unlikely, though conceivable, that I missed something that should affect my ratings.

---

> ### Author Rebuttal · Authors · 2023-08-25
>
> We thank you for your comments, and agree that our evaluation setup can inform further work examining the relationship between segmentation and lexical access.
>
> To the best of our knowledge, our methodology and metric are novel. If you could point us to prior work evaluating subword tokenizers through cognitive task performance, we would greatly appreciate it.

---

### Meta-Review · Area_Chair_Sqac · 2023-09-19

**Recommendation:** 5

**Metareview:**

The content and results are clear and a great fit for this track. The focus on cognitive aspects is particularly relevant.

---

### Decision · Program_Chairs · 2023-10-07

**Decision:**

Accept-Main

**Comment:**

The content and results are clear and a great fit for this track. The focus on cognitive aspects is particularly relevant.